# Pyrolysis Temperature vs. Application Rate of Biochar Amendments: Impacts on Soil Microbiota and Metribuzin Degradation

**DOI:** 10.3390/ijms241311154

**Published:** 2023-07-06

**Authors:** Kamila Cabral Mielke, Maura Gabriela da Silva Brochado, Ana Flávia Souza Laube, Tiago Guimarães, Bruna Aparecida de Paula Medeiros, Kassio Ferreira Mendes

**Affiliations:** 1Department of Agronomy, Federal University of Viçosa, Viçosa 36570-900, MG, Brazil; kamila.mielke@ufv.br (K.C.M.); maura.brochado@ufv.br (M.G.d.S.B.); bruna.medeiros@ufv.br (B.A.d.P.M.); 2Department of Chemistry, Federal University of Viçosa, Viçosa 36570-900, MG, Brazil; anaf.laube@gmail.com (A.F.S.L.); tguimaraes.quimica@gmail.com (T.G.)

**Keywords:** application rate, carbonaceous material, degradation time, herbicide residual, pyrolysis temperature

## Abstract

Biochar-amended soils influence the degradation of herbicides depending on the pyrolysis temperature, application rate, and feedstock used. The objective of this study was to evaluate the influence of sugarcane straw biochar (BC) produced at different pyrolysis temperatures (350 °C, 550 °C, and 750 °C) and application rates in soil (0, 0.1, 0.5, 1, 1.5, 5, and 10% *w*/*w*) on metribuzin degradation and soil microbiota. Detection analysis of metribuzin in the soil to find time for 50% and 90% metribuzin degradation (DT_50_ and DT_90_) was performed using high-performance liquid chromatography (HPLC). Soil microbiota was analyzed by respiration rate (C-CO_2_), microbial biomass carbon (MBC), and metabolic quotient (*q*CO_2_). BC350 °C-amended soil at 10% increased the DT_50_ of metribuzin from 7.35 days to 17.32 days compared to the unamended soil. Lower application rates (0.1% to 1.5%) of BC550 °C and BC750 °C decreased the DT_50_ of metribuzin to ~4.05 and ~5.41 days, respectively. BC350 °C-amended soil at high application rates (5% and 10%) provided high C-CO_2_, low MBC fixation, and high *q*CO_2_. The addition of low application rates (0.1% to 1.5%) of sugarcane straw biochar produced at high temperatures (BC550 °C and BC750 °C) resulted in increased metribuzin degradation and may influence the residual effect of the herbicide and weed control efficiency.

## 1. Introduction

Herbicides are used for weed control in pre- or post-emergence, and regardless of the mode of application, they can reach the soil and persist with a residual effect for weed control and cause carryover problems in succeeding crops or contaminate non-target organisms and the environment [1]. The negative impacts of herbicide residues can be categorized based on the chemical structure of the herbicides, crop species, environmental conditions, and soil properties. These impacts include plant phytotoxicity, reduction in biomass with or without recovery, and significant impairment of crop development [2]. Contamination of potable and groundwater, soil resources, and microbial activities are considered critical aspects regarding the environmental risks associated with herbicide use [3]. Furthermore, the effects of herbicides on human health, particularly due to the bioaccumulation of these substances’ molecules in the body, represent a significant concern in terms of herbicide-related biosafety [4].

The soil is the main site where physical, chemical, and biological interactions of herbicides occur [5]. Herbicide degradation into secondary compounds (metabolites) can occur through biotic processes (microbial degradation) or abiotic processes (hydrolysis, photolysis, and oxidation) [6,7]. Hydrolysis is characterized by the breakdown of the herbicide molecule through hydrolysis reactions involving ether, amide, cyano group, and acyl chloride bonds. In the photolysis reaction, the herbicide absorbs light radiation and generates hydroxyl radicals, superoxide, and ozone, which induce a molecular reaction, breaking the bonds and degrading the herbicide [8]. The rate of herbicide degradation depends on the type of soil, pH, organic carbon (OC) content, moisture, and soil colloid nature [9].

Biochar is a carbonaceous material produced during the thermochemical decomposition (pyrolysis) of biomass under a limited O_2_ supply [10]. The pyrolyzed feedstocks and pyrolysis conditions determine the physicochemical characteristics of biochar, such as nutrient content, OC, porosity, and specific surface area (SSA), among others, which are determinants for herbicide sorption and degradation [11]. The application of biochar generally stimulates the establishment of local microbial communities, such as arbuscular mycorrhizal fungi and bacteria, due to the increased availability of OC and nutrients in the soil [12,13].

Biochar-amended soils can directly influence the degradation half-life time (DT_50_) of herbicide depending on the molecule, pyrolysis temperature, application rate, and feedstock used [1]. For example, the total amount of hexazinone (mineralized residue + non-extracted or bound residue) in soils amended with biochar from eucalyptus wood waste produced at pyrolysis temperatures of BC850 °C (46%) and BC950 °C (49%) was higher compared to biochar produced at BC650 °C (33%) and BC750 °C (42%) [14]. In contrast, the degradation of the non-ionizable herbicide oxyfluorfen applied in pre-emergence was faster (DT_50_ of 2 days and 23 days) with the addition of rice husk biochar produced at 500 °C, at an application rate of 2%, than in unamended soil [15]. Different application rates of hardwood biochar showed positive effects on the mineralization of ^14^C-atrazine in Brazilian soil, representing increases of 50% (0.1% *w*/*w*), 48% (1.0% *w*/*w*), and 46% (5.0% *w*/*w*) compared to unamended soil [16]. The high persistence of herbicides in soils amended with biochar, due to the unavailability of the molecules in the soil solution for microorganisms, can significantly alter biodegradation processes; however, degradation responses are dependent on the physicochemical characteristics of the herbicide, biochar production, and soil type [1,17].

The relationship between the addition of biochar to soil and the degradation of metribuzin is reported in the scientific literature; however, the results are distinct. Metribuzin is a pre- and post-emergent herbicide, a selective residual of the triazinone group, with the ability to effectively control a wide spectrum of eudicot weeds [18]. The herbicide blocks photosynthetic processes by inhibiting electron transport in photosystem II (PSII), causing physiological and morphological changes in leaf structures that undergo necrosis and death [19]. Metribuzin [4-amino-6-tert-butyl-3-methylsulfanyl-1,2,4-triazin-5-one] is a strong acid herbicide with high water solubility (*S*_w_ = 10.700 mg L^−1^ at 20 °C), high mobility in soil (coefficient of sorption normalized by OC, *K*_oc_ = 38 mg L^−1^), high leaching index (groundwater ubiquity score = 2.96), and low persistence in soil (DT_50_ = ~20 days) [20,21,22]. This herbicide showed high sorption in carbonaceous materials produced at different pyrolysis temperatures [13,23,24]. The addition of biochar and organic compost decreased the DT_50_ of metribuzin by 4 days when compared to the unamended soil (DT_50_ = 34 days) [25]. The degradation and mobility of metribuzin under simulated light and dark conditions decreased with the addition of 1% biochar [26]. The authors reported that can be attributed to the enhanced sorption properties of biochar.

To understand the impact of biochar on the fate of metribuzin in soil, it is necessary to detail the effects of different types of biochar, pyrolysis temperature, and application rate on metribuzin degradation in soil, which is important to assess the risk and fate of this herbicide in the environment. Therefore, the objective of this study was to evaluate the influence of biochar from sugarcane straw produced at different pyrolysis temperatures and application rates in soil on metribuzin degradation and soil microbiota.

## 2. Results

### 2.1. Metribuzin Degradation

The sugarcane straw biochar (BC) samples produced at different pyrolysis temperatures (350 °C, 550 °C and 750 °C) were denominated BC350 °C, BC550 °C and BC750 °C. The interaction between pyrolysis temperature, application rate of biochar in the soil, and degradation time was significant (*F* = 2.738 and *p* ≤ 0.002). The regression curves of metribuzin over time in amended and unamended soils with different application rates of biochar (0, 0.1, 0.5, 1, 1.5, 5, and 10% *w*/*w*) produced at different pyrolysis temperatures (BC350 °C, BC550 °C, and BC750 °C) are presented in Figure 1. The first-order kinetic model provided a suitable fit with metribuzin degradation with a coefficient of determination (*R*^2^) greater than 0.940 (Table 1). The degradation time values of 50% (DT_50_) and 90% (DT_90_) of metribuzin applied in the unamended soil were 7.37 days and 24.49 days, respectively (Table 1). The temperature and rainfall during the study varied from 25 °C to 35 °C and from 0 mm to 4 mm, respectively (Figure 2). At 150 days after application (DAA), the degradation of metribuzin in the unamended soil was approximately 100% of the initially applied amount.

The intensity of metribuzin degradation was dependent on the pyrolysis temperature and application rate of biochar in the soil (Figure 1 and Table 1). At 150 DAA, metribuzin degradation in amended soil with BC350 °C, BC550 °C, and BC750 °C was approximately 100% of the initially applied concentration. The initial concentration (*C*_0_) of metribuzin in amended soil with BC350 °C, BC550 °C, and BC750 °C varied among the application rates (Figure 1 and Table 1).

BC350 °C-amended soil increased the DT_50_ of metribuzin from 7.35 days to 17.32 days and the DT_90_ from 24.41 days to 57.56 days compared to unamended soil, when application rates when the application rates were 0.5%, 1, 5% and 10%. BC550 °C-amended soil increased the DT_50_ of metribuzin from 3.14 days to 6.08 days as the application rate increased from 0.1% to 10%. The DT_50_ of metribuzin for BC750 °C was similar among the application rates ~6.59 days, except for the 0.5% and 1% rates, which were below this value (Table 1).

The application rates of biochar produced at different pyrolysis temperatures influenced the DT_50_ of metribuzin in soil (Table 1). Application rates of 0.1%, 0.5%, and 1.0% of BC350 °C demonstrated metribuzin DT_50_ similar to unamended soil (7.37 days). Metribuzin DT_50_ increased from 7.37 days in unamended soil to 9.62 days and 17.32 days when rates of 5% and 10% of BC350 °C were added, respectively. Regardless of the application rate of BC550 °C and BC750 °C, the DT_50_ of metribuzin was lower than in unamended soil. However, it was observed that with an increase in the application rate from 0.1% to 10% of BC350 °C and BC550 °C, the DT_50_ increased on average by 3 days. Lower application rates of biochar produced at different pyrolysis temperatures decreased the DT_50_ and DT_90_ of metribuzin, aligned mainly with the higher pyrolysis temperatures (BC550 °C and BC750 °C).

### 2.2. Respiration Rate of the Microbial Rhizosphere

The timing of metribuzin application in biochar-amended soil produced at different pyrolysis temperatures and application rates showed different responses in soil microbiological parameters (Figure 3 and Figure 4, Table 2). The maximum respiration rate of unamended soil was at 9 days after incubation (DAI), with values of 600 and 800 µg CO_2_ g soil^−1^ day^−1^, respectively, at 0 DAA and 150 DAA of metribuzin, respectively. The maximum respiration rate at 0 DAA was at 9 DAI, with respiration ranging from 600 µg CO_2_ g soil^−1^ day^−1^ to 900 µg CO_2_ g soil^−1^ day^−1^, depending on the application rate and pyrolysis temperature. At 150 DAA, the maximum respiration was between 9 and 13 DAI, with respiration rates between 400 µg CO_2_ g soil^−1^ day^−1^ and 900 µg CO_2_ g soil^−1^ day^−1^, depending on the application rate of BC350 °C, BC550 °C, and BC750 °C (Figure 3).

The interaction between pyrolysis temperature and biochar soil application rate for total respiration was significant for 0 DAA (*F* = 2.796 and *p* ≤ 0.03) and for 150 DAA (*F* = 3.140 and *p* ≤ 0.01). The unamended soil presented a total respiration (C-CO_2_) of 3066 µg CO_2_ g soil^−1^ and 2503 µg CO_2_ g soil^−1^ at 0 DAA and 150 DAA, respectively (Figure 4). At 0 DAA, C-CO_2_ was influenced by the pyrolysis temperature only for the highest application rates (5% and 10%) of BC350 °C, BC550 °C, and BC750 °C. In BC350 °C-amended soil, higher C-CO_2_ was observed by 20.7% (average of rates 5% and 10%) compared to BC550 °C and BC750 °C-amended soil. There was no significant effect between application rates of BC550 °C and BC750 °C in the soil. The application rates of 5% and 10% of BC350 °C provided greater C-CO_2_ from the soil microbiota, 4424 µg CO_2_ g soil^−1^ and 4644 µg CO_2_ g soil^−1^, respectively, when compared to unamended soil (3066 µg CO_2_ g soil^−1^) (Figure 4). There were no differences in C-CO_2_ for different application rates when BC750 °C was used at 150 DAA (Figure 4). Rates of 5% and 10% showed C-CO_2_ of 2933 µg CO_2_ g soil^−1^ and 3072 µg CO_2_ g soil^−1^, respectively, when BC350 °C was added. Rates between 1% and 10% of BC550 °C increased C-CO_2_, on average, by 13% compared to rates of 0.1% and 0.5% (Figure 4).

The interaction between factors for MBC was significant at 0 DAA (*F* = 8.565 and *p* ≤ 0.001) and for 150 DAA (*F* = 5.549 and *p* ≤ 0.001). Unamended soil presented an average MBC of 40 µg MBC g soil^−1^ and 60 µg MBC g soil^−1^ at 0 DAA and 150 DAA, respectively (Figure 4). At 0 DAA, BC350 °C and BC550 °C-amended soil increased MBC by > 100% when compared to BC750 °C at a 1.0% application rate. Application rates of 0.5%, 1.0%, and 1.5% provided higher MBC, averaging 152 µg MBC g soil^−1^, for BC350 °C compared to other application rates. Rates of 0.1%, 0.5%, and 1.0% increased MBC by an average of 67.0% compared to rates of 0%, 1.5%, 5.0%, and 10% for BC550 °C. The 0.5% rate of BC750 °C-amended soil increased MBC by 65% when compared to unamended soil and rates of 0.1%, 1.0%, 1.5%, 5.0%, and 10% (Figure 4). At 150 DAA, BC750 °C and BC550 °C-amended soil reported higher MBC, averaging 31.7%, when compared to BC350 °C at a 10% rate (Figure 4). The 5% rate of BC350 °C showed higher MBC (225 µg MBC g soil^−1^) when compared to unamended soil and rates of 0.1%, 0.5%, 1.0%, 1.5%, and 10% (averaging 131.87 µg MBC g soil^−1^). Rates of 5% and 10% of BC550 °C-amended soil increased MBC by 42.2%, on average, when compared to unamended soil, and rates of 0.1%, 0.5%, 1.0%, and 1.5% (Figure 4). 

The interaction between factors for metabolic quotient (*q*CO_2_) was significant at 0 DAA (*F* = 7.727 and *p* ≤ 0.001) and for 150 DAA (*F* = 6.914 and *p* ≤ 0.003). The metabolic quotient (*q*CO_2_) of unamended soil at 0 DAA and 150 DAA was 63.7 µg CO_2_/µg MBC and 39.1 µg CO_2_/µg MBC, respectively (Figure 4). At 0 DAA, BC750 °C-amended soil showed high *q*CO_2_ (73.9 µg CO_2_/µg MBC) when compared to BC550 °C and BC350 °C-amended soil (average of 21.49 µg CO_2_/µg MBC) at a rate of 1.0%. Rates of up to 5.0% application of BC350 °C showed low *q*CO_2_ (average of 23.77 µg CO_2_/µg MBC) when compared to unamended soil and a rate of 10% (average of 63.7 µg CO_2_/µg MBC). The highest rates (1.5%, 5.0%, and 10%) of BC550° and BC750 °C provided *q*CO2 ~ 58.4 µg CO_2_/µg MBC, a value lower than unamended soil (63.7 µg CO_2_/µg MBC) (Figure 4). At 150 DAA, BC750 °C-amended soil showed, on average, higher *q*CO_2_ (11.1 µg CO_2_/µg MBC) when compared to BC550 °C and BC350 °C-amended soil (Figure 4). Application rates of BC350 °C, BC550 °C, and BC750 °C showed significant results only in relation to unamended soil. The application rates provided an average *q*CO_2_ of 17.6, 16.4, and 19.3 µg CO_2_/µg MBC for BC350 °C, BC550 °C, and BC750 °C-amended soil, respectively (Figure 4).

## 3. Discussion

The values of metribuzin DT_50_ range from 7.03 days to 138 days, depending on climate, soil, and field and laboratory conditions [7,22,27,28,29]. Generally, its degradation occurs more rapidly in the first month after application. In the conditions of this study, temperature and humidity were tropical, with high temperatures and well-distributed rainfall, using soil with an OC content of 1.2% and a sandy loam texture. These characteristics may have favored the rapid degradation of metribuzin during the study, as the DT_50_ values of metribuzin were dependent on temperature, humidity, and soil physicochemical characteristics, such as OC content [21,30,31]. Metribuzin degradation showed a strong dependence on soil type (sandy or silty) and temperature (5 °C, 15 °C, and 28 °C) [30]. The authors reported that an increase in temperature from 5°C to 15 °C reduced the concentration of metribuzin in the soil by 25%, decreasing the DT_50_ from ~385 days to 105 days for soil with a higher sand content (~60%) and lower OC content (0.15%). Metribuzin has high *S*_w_, low sorption, and persistence, suggesting a high potential for movement and ready availability for microbial degradation in the soil. This is particularly true for soils with high OC content [32,33]. The impact of abiotic factors such as soil type, application rate, soil pH, microorganisms, and sunlight on metribuzin persistence has been evaluated [34]. The authors observed that DT_50_ values varied significantly with the metribuzin application rate and the physical–chemical characteristics of the soil, with values ranging from 15.17 days to 46.59 days.

The variation in *C*_0_ may have occurred due to the initial sorption process of metribuzin in soils with different application rates, which directly reflects the amount of bioavailable herbicide in the soil solution for degradation. Mielke et al. [18] reported that high doses of metribuzin in soil (>2 mg L^−1^) were less sorbed (60%) in soils amended with BC350 °C, especially for lower application rates (0.1% to 1.5%). The authors reported that the amended soil with BC550 °C and BC750 °C showed sorption percentages greater than 80% of metribuzin in the soil. Biochars have different physical and chemical characteristics as the pyrolysis temperature changes during production. Herbicide sorption is directly influenced by characteristics such as porosity, SSA, aromatic structures, carbon contents, surface functional groups, pH, and the elemental composition of the soil [35]. The biochar produced at 750 °C (BC750 °C) showed a higher C/N ratio, ash content, lower number of surface functional groups, and a 13-fold higher surface area than the biochar produced at 350 °C (BC350 °C), which increases the sorption capacity of BC750 °C [24].

The results showed that biochar produced at low temperatures (BC350 °C) negatively influenced the degradation of metribuzin in the soil. The greater degradation of metribuzin in soils amended with BC550 °C and BC750 °C may be related to the physicochemical characteristics of the biochar, as high pyrolysis temperatures produced a material with high pH (9.7), ash content (11%), OC (1.4%), and high nutrient content [24]. These factors can favor chemical degradation through increased pH, active groups, and generation of free radicals [36], and the level of C and ash content can affect soil microbial activities due to the presence of nutrient-rich materials (N, S, P, among others). Soil modifications with BC300 °C from maize straw promoted the biological degradation of triclopyr, increasing the abundance of microorganisms and improving the activity of nitrile hydratase (NHase) [36]. The authors reported that modifications with BC500 °C and BC700 °C inhibited biodegradation by reducing the availability of triclopyr; however, chemical degradation occurred mainly through high pH, active groups on the mineral surface, and generation of hydroxyls and other free radicals.

The effect of biochar on the degradation behavior of herbicides in the soil is a complex process that involves the interaction between soil, herbicide, and the physicochemical characteristics of biochar. When an adequate amount of biochar was added to the soil (0.1% to 1.5%), better degradation of metribuzin was observed, which may be related to the inhibition of microbial activity by excess biochar, influencing the richness and abundance of the microbial community and the higher sorption capacity of the herbicide at high rates of application of sugarcane straw biochar produced at high pyrolysis temperatures (>500 °C) [24,26,37,38]. Therefore, as the sorption rate increased, the DT_50_ of metribuzin also increased. In most cases, degradation decreases with increasing biochar application rates, associated with greater herbicide sorption [39,40,41]. Similar results were observed in the degradation of flumioxazin in soils with different biochar contents [42]. The authors found that soil amended with cornstalk biochar with application rates of 0, 0.5, 2.0, 5.0, and 10.0% provided a DT_50_ of flumioxazin of 11.1, 9.0, 11.1, 13.2, and 15.4 days, respectively.

Metribuzin is a strong acid (acid dissociation constant, p*K*a = 1.3), and under conditions of high pH, it is more in the ionic form, negatively charged, increasing the repulsion of soil colloids, mainly the predominance of negative charges in the organic matter [43,44]. The soil pH with BC350 °C addition was lower (8.6) than BC550 °C (9.3) and BC750 °C (9.8), and higher pH values were observed for soil corrected with BC750 °C, where doses of 5% and 10% increased pH by 1.7 and 2.1 units, respectively, compared to uncorrected soil [24]. Therefore, under these conditions, a greater amount of metribuzin was available for microbial degradation in the soil, which may explain why the DT_50_ value of metribuzin was lower than in unamended soil, regardless of the application rate of BC550 °C and BC750 °C. Although Mielke et al. [24] reported greater sorption at higher application rates of BC550 °C and BC750 °C, these materials showed greater potential for microbial colonization and, consequently, greater degradation of metribuzin.

Biological degradation and mineralization are the main pathways of herbicide dissipation in the soil [45]. When herbicides are sorbed onto biochar, their availability for degradation by microorganisms is reduced. However, sorption can be reversed by the process of desorption, and any subsequent remobilization of the herbicide, if bound as a residue to biochar, can create availability for further degradation in the soil solution. The addition of BC550 °C and BC750 °C at low application rates in the soil (0.1% to 1%) increased metribuzin sorption [24] compared to unamended soil and resulted in greater metribuzin degradation, potentially acting initially as a remediation technique to immobilize herbicides via sorption, and eventually desorbing the molecules from soil colloids. Consequently, the combination of increased sorption and increased biological degradation has the potential to effectively reduce metribuzin in the soil. However, this positive effect on the remediation of contaminated soils may negatively influence the agronomic efficacy of metribuzin in the soil, reducing the control of weed seed banks. The fact that biochar can reduce herbicide efficacy is not a desired effect, as it may require an increased herbicide dose to achieve similar levels of control as in soil not altered by biochar, increasing production costs and potential environmental risk [46]. Therefore, it is important to understand the physicochemical characteristics of biochar, soil, and herbicides, as well as the possible interactions between these factors, in order to achieve satisfactory recommendations and functionality of biochars as fertilizer sources and herbicide sorbents without posing an environmental risk.

The pyrolysis temperature and application rate of biochar influenced the soil microbiota. In general, structural and compositional differences in soil microbiota can be related to biochar (feedstock, pyrolysis condition, application rate), soil (pH, OC, temperature, moisture, aeration), environmental factors (vegetation, land use, management intensity, herbicide action), and physicochemical characteristics of herbicides [47,48]. In the present study, biochar mitigated the negative impact of metribuzin on soil microbial community, and these effects may be related to increased microbiota through interaction mechanisms with biochar, such as physical-chemical structure (macro and micropores, surface area, nutrient content, organic substances, and enzymatic activity) and increased sorption, reducing bioavailability and toxicity to soil microbiota [24,49,50,51,52]. In the study by Mielke et al. [24], it was reported that the use of application rates of 1% and 1.5% of BC350 °C, BC550 °C, and BC750 °C improved soil fertility, making P, K, Mg, Fe, and Mn available, reducing potential acidity (H + Al), and increasing soil pH with less impact on metribuzin sorption and desorption. These soil modifications possibly had a positive impact on the increased degradation of metribuzin in soil amended with BC550 °C and BC750 °C, as observed in this study.

Microorganisms can use hydrocarbons on the surface of biochar as a carbon source [53]. Organic-mineral complexes can form on the surface and in the pores of biochar, modifying its sorption properties and providing habitats for microbial colonization [44,54]. Biochar can provide habitats for different microorganisms through its pores of different sizes (macro-, meso-, and micropores), which potentially protect these microorganisms from desiccation and predation [55]. Furthermore, biochar application can increase the availability of mineral elements and microbiological activity, which may be related to cation exchange capacity (CEC) and soil pH increase [44]. The increase in soil pH with biochar addition can reduce aluminum toxicity and increase nutrient availability, generating positive effects on microbial colonization [56].

Higher basal respiration found in treatments with higher pyrolysis temperatures and biochar application rates may be an indication of increased biological activity in these treatments. However, the increase in microbial basal respiration must be aligned with the increase in MBC since a high respiration rate and low MBC indicate negative changes in soil microbiota [57]. High *q*CO_2_ values suggest unfavorable conditions for soil microbiota, and low values indicate greater MBC efficiency [58,59]. BC350 °C-amended soil at high application rates provided a microbial imbalance in the soil, as it showed high respiration, low microbial carbon fixation, and high metabolic quotients, unlike BC550 °C and BC750 °C-amended soil (Figure 4). These results are consistent with those observed in the study of metribuzin degradation in soil, where a higher DT_50_ value of metribuzin was observed in BC350 °C-amended soil when compared to BC550 °C and BC750 °C (Table 1).

The addition of a large amount of carbon can stimulate enhanced microbial action in the soil and therefore cause greater microbial degradation. In addition, the addition of biochar produced at high pyrolysis temperatures increased soil nutrient levels, especially P, K, and SSA (223 m^2^ g^−1^) [24], which can stimulate microbial activity and consequently improve biological degradation. The increase in MBC can also be attributed to the addition of biochar produced at high temperatures, which constituted the most readily available source of energy for soil microorganisms. Microbial biomass is considered the living fraction of soil organic matter (OM) and a significant nutrient reservoir [60]. The application of hardwood-derived biochar increased the mineralization of atrazine by stimulating atrazine-adapted microflora compared to unamended soil [16].

At 150 DAA, it was possible to observe that the *q*CO_2_ of soils, regardless of pyrolysis temperature and application rate, were lower than the unamended soil. Possibly, the addition of biochar in the soil boosted the microbiota, reducing the energy expenditure used in maintaining the microbial community and directing resources to cell synthesis, improving microbial growth [61].

Biochar has a high capacity for sorbing and retaining soluble organic matter, gases, nutrients, and water and can therefore provide soil microorganisms with various energy resources, nutrients, moisture, and the formation of macroaggregates [37,62,63,64]. Although biochar is highly recalcitrant, it can be degraded by microorganisms co-metabolically [65]. The labile part of biochar is biologically degradable in a few months after application, while the stable fraction consists of recalcitrant compounds that remain years after biochar application [66]. Increases in nutrients and labile C can be provided by biochar application to soil, and the effect of herbicide mineralization will depend on the proportion of labile C and nutrient content in the applied biochar. Therefore, the impact of biochar application on soil is dependent on the physicochemical characteristics of biochar, which may differ when the amount applied or the raw material source is altered. Two biochars produced from different feedstocks (cocoa husk and rice husk) applied at a rate of 0.3% were analyzed for the degradation of atrazine and paraquat in soil [67]. The authors reported that cocoa husk biochar increased MBC by an average of 72% for atrazine and paraquat compared to rice husk biochar. This was due to the higher level of the nutrient composition of total N and available P in cocoa husk biochar compared to rice husk. The higher degradation of oxyfluorfen was observed in soils amended with different rates of rice husk biochar application, decreasing DT_50_ between 2 days and 23 days compared to unamended soil [68].

In addition, sugarcane straw biochar produced at high pyrolysis temperatures (>550 °C) and at moderate application rates (between 0.5% and 1.5%) boosted the soil microbiota and improved metribuzin degradation. This result reinforces the idea that materials with high sorptive capacity, such as biochar produced at high pyrolysis temperatures, can be used as soil amendments to improve soil microbiota as long as the application rate is controlled. However, it is important to note that this result may differ in other soil types, biochar feedstocks, application rates, and pyrolysis temperatures, and specific studies are necessary for each utilization scenario.

## 4. Materials and Methods

### 4.1. Soil Collection and Analysis

The agricultural soil samples were collected from the top layer (0–10 cm) in Viçosa, MG, Brazil (20°46′05″ S; 42°52′08″ W), an area that has not been treated with herbicides for the last three years. The soil samples were air-dried for 10 days, then sieved on 5.0 mm mesh and stored at room temperature. The soil was classified as Oxisol (*Latossolo Vermelho-Amarelo*).

The sugarcane straw biochar (BC) samples produced at different pyrolysis temperatures (350 °C, 550 °C, and 750 °C) were denominated BC350 °C, BC550 °C and BC750 °C. The soil was amended with sugarcane straw biochar produced at different pyrolysis temperatures (BC350 °C, BC550 °C, and BC750 °C) in the application rates of 0, 0.1, 0.5, 1, 1.5, 5, and 10% (*w*/*w*) representing 0, 1, 5, 10, 15, 50, and 100 Mg ha^−1^, respectively, assuming a soil density of 1 g cm^−3^ and incorporation depth of 0.10 m. The physicochemical attributes of unamended and biochar-amended soil produced at different pyrolysis temperatures were reported by Mielke et al. [24], shown in Table 2.

### 4.2. Sugarcane Straw Biochar

The sugarcane straw was crushed, sieved (10 mesh, <2.0 mm), and dried in an oven with forced air circulation at 103 ± 2 °C for 72 h. The straw was placed in a sealed reactor to prevent the ingress of O_2_. The reactor oven was heated at a rate of 5 °C min^−1^, and the pyrolysis temperatures were 350 °C, 550 °C, and 750 °C. The physicochemical characterization of sugarcane straw biochar was described by Mielke et al. [24], shown in Table 3.

### 4.3. Soil Preparation and Application of Metribuzin

The experimental design was a completely randomized triple factorial scheme 3 × 7 × 10 with 3 replications. The first factor was three pyrolysis temperatures (350 °C, 550 °C, and 750 °C), the second factor was the application rates of biochar in the soil (0, 0.1, 0.5, 1.0, 1.5, 5.0, and 10% (*w*/*w*)), and the third factor was the evaluation time (0, 5, 10, 15, 30, 45, 60, 90, 120, and 150 days). The soil amended with sugarcane straw biochar was added to pots with a capacity of 0.5 kg. The application of metribuzin (Sencor^®^480, Bayer CropScience LP, Kansas City, MO, USA) was carried out at the maximum recommended dose (1920 g a.i. ha^−1^) for sugarcane crop, with a control treatment without herbicide application. In this procedure, a CO_2_-pressurized sprayer equipped with two TT110.02 nozzles spaced 0.5 m apart was used, maintained at a pressure of 1.96 bar and a spray volume of 170 L ha^−1^. The pots were kept in a greenhouse, and soil samples were collected at 0, 5, 10, 15, 30, 45, 60, 90, 120, and 150 days after herbicide application (DAA). The temperature inside the greenhouse was recorded during the experiment. Soil moisture was adjusted by irrigation of the pots according to the rainfall distribution in 2021 for Viçosa, MG, Brazil [69] (Figure 2).

At the time of collection, the soil amended with biochar was homogenized, and the samples were stored in previously identified jars and taken to freezing at −18 °C in a freezer for later chromatographic analysis.

### 4.4. Extraction of Metribuzin

The stock solution was prepared at a concentration of 500 mg L^−1^ of the analytical standard Metribuzin-Pestanal™ (98.8% purity Sigma-Aldrich, San Luis, MO, USA) and the working solution at a concentration of 100 mg L^−1^, both in acetonitrile (99.9% purity grade). From the working solution, three concentrations of metribuzin (2.45, 3.45, and 4.45 mg L^−1^) were prepared. The extraction of the herbicide in the soil was performed as described by Mehdizadeh et al. [70]. The method consisted of adding 20 mL of the extraction solution (methanol) to Falcon tubes containing 5 g of soil. Then, the tubes were subjected to rotary shaking for 24 h [13] and centrifuged (Kasvi, K14-0815P, Curitiba, Paraná, Brazil) at 1372× *g* for 7 min. The supernatant was collected and filtered through a Millipore filter (PRFE membrane, 0.45 µm). An aliquot of 1.50 mL was placed in a vial to be analyzed in high-performance liquid chromatography (HPLC, LC 20AT, Shimadzu, Kyoto, Japan). The recovery level of metribuzin in fortified soil samples was, on average, 100.2%.

### 4.5. Chromatographic Conditions

Validation of the chromatographic method was according to the criteria of ANVISA [71]. The linearity of the extraction method was determined by preparing analytical curves where soil samples were fortified with different concentrations of metribuzin (0.01, 0.05, 1.00, 1.50, 2.45, 3.45, 4.45, 5.00, and 5.45 mg kg^−1^). After chromatographic analysis and obtaining the analytical curves, linearity was evaluated by linear regression of the area as a function of metribuzin concentrations and the coefficient of determination (*R²*). The analytical curve presented an *R*^2^ equal to 0.9999 (Figure 5). The limit of detection (LoD) and quantification (LoQ) were 0.01369 and 0.04150 mg L^−1^, respectively. 

The quantification of metribuzin was carried out on an HPLC, with a photodiode array detector (SPD-M20A, Shimadzu, Kyoto, Japan), stainless steel C18 column (Shimadzu VP-ODS Shim-pack 250 mm × 4.6 mm i.d., 5 µm particle size, Shimadzu, Kyoto, Japan). The mobile phase was adapted from López-Piñeiro et al. [13,24], composed of acetonitrile/water (acidified with 0.01% phosphoric acid) in a ratio of 45/55 (*v*/*v*), an injection volume of 30 µL, flow rate of 1.0 mL min^−1^, wavelength of 254 nm, and column oven temperature of 30 °C. Under these conditions, the retention time was 8.2 min.

### 4.6. Degradation Kinetics of Metribuzin in Soil

The degradation data of metribuzin in the unamended soil and biochar-amended soil were fitted to a first-order kinetics model according to Equation (1).
(1)$$C_{t}=C_{0}\times e^{-kxt}$$
where *C_t_* is the total concentration (mg kg^−1^) of herbicide remaining in the soil at time *t*; *C*_0_ is the initial concentration of herbicide at time zero; *k* is the degradation rate constant (days^−1^), and *t* is incubation time in days.

From the *k* values, the time required for 50% and 90% of the initial amount of metribuzin to be degraded (DT_50_ and DT_90_) was determined (Equations (2) and (3)).
(2)$${DT}_{50}=\frac{In2}{k}$$
(3)$${DT}_{90}=\frac{In10}{k}$$

### 4.7. Respiration Rate of the Microbial Rhizosphere

After completion of the collections of the unamended and biochar-amended soil for analysis of metribuzin degradation at 150 DAA and 0 DAA (at the same time as herbicide application), the carbon content in soil microbial biomass by induced respiration was based on the measurement of the initial maximum emission of CO_2_ over a period of time [72]. In this study, an entirely randomized design was performed in a 3 × 7 double factorial scheme with 4 repetitions. The first factor represented the pyrolysis temperatures (BC350 °C, BC550 °C, and BC750 °C), and the second factor was the biochar soil application rates (0, 0.1, 0.5, 1.0, 1.5, 5.0, and 10% *w*/*w*). 

The soil collected from the rhizosphere of each experimental unit was homogenized, and 50 g was taken for analysis. Soil samples were sieved (2 mm mesh), moistened (70% field capacity), and incubated in hermetically sealed vials in a Biochemical Oxygen Demand chamber (BOD ElectroLab, São Paulo, Brazil) at 25 °C without light. The respiratory frequency of the soil microbiota was evaluated with the respirometric method of C-CO_2_ release at 3, 6, 9, 13, 17, 22, 27, and 35 days after the start of incubation (DAI). The C-CO_2_ released from the soil was transported by a continuous air flow (CO_2_-free) to a vial containing 10 mL of 0.5 mol L^−1^ NaOH solution. Precipitation of the carbonate formed was carried out with the addition of 10 mL of BaCl_2_ 0.05 mol L^−1^ and titrated with 0.25 mol L^−1^ HCl plus three drops of the 1% phenolphthalein indicator [73]. After 40 days, 10 g soil samples from each vial were taken to determine the microbial biomass carbon (MBC) [74]. The metabolic quotient (*q*CO_2_) was determined as follows in Equation (4) [75].
(4)$${qCO}_{2}=\frac{{C-CO}_{2}}{MBC}$$

### 4.8. Statistical Analysis

The data were subjected to analysis of variance (ANOVA) to evaluate the interaction between factors in each study. The analyses were performed using Sisvar software (version 5.6, Lavras, Minas Gerais, Brazil). When the interaction between factors was significant (*p* ≤ 0.05), the degradation curves of metribuzin in soil were plotted in Sigma Plot^®^ (version 14.0 for Windows, Systat Software Inc., Point Richmond, VA, USA), and the parameter data were presented as means and standard deviation (*n* = 3). For C-CO_2_, MBC, and *q*CO_2_, when significant (*p* ≤ 0.05) among factors, the means were separated by Tukey’s test and presented as means and standard deviation (*n* = 4), and the figures were also plotted in Sigma Plot^®^.

## 5. Conclusions

The degradation values of metribuzin (DT_50_ and DT_90_) in unamended soil were 7.37 days and 24.94 days, respectively, reflecting the low residual effect capacity of this herbicide in the studied soil.

The intensity of metribuzin degradation was dependent on the pyrolysis temperature and biochar application rate in the soil. The highest degradation of metribuzin was observed in soils amended with BC550 °C and BC750 °C when added at lower application rates (0.1% to 1.5%). The highest values of DT_50_ and DT_90_ for metribuzin were observed in BC350 °C-amended soil applied at rates of 5% (9.62 days) and 10% (17.32 days).

The degradation process of metribuzin in BC350 °C, BC550 °C, and BC750 °C was shown to be related to the negative impact of these carbonaceous materials on the soil microbiota since in BC350 °C-amended soil at high application rates, a higher microbial imbalance was observed, presenting high respiration, low microbial carbon fixation, and high metabolic quotients, unlike BC550 °C and BC750 °C-amended soil. Even though providing greater sorption of metribuzin in the soil, the addition of low application rates of BC550 °C and BC750 °C to the soil may lead to an increase in metribuzin degradation.

This material can also become an alternative for environmental remediation and reduce problems related to crop carryover. However, it may negatively influence the residual effect of the herbicide in the soil and consequently reduce the efficacy of the product in controlling weeds in the seed bank and increase the application of post-emergent herbicides.

## Figures and Tables

**Figure 1 ijms-24-11154-f001:**
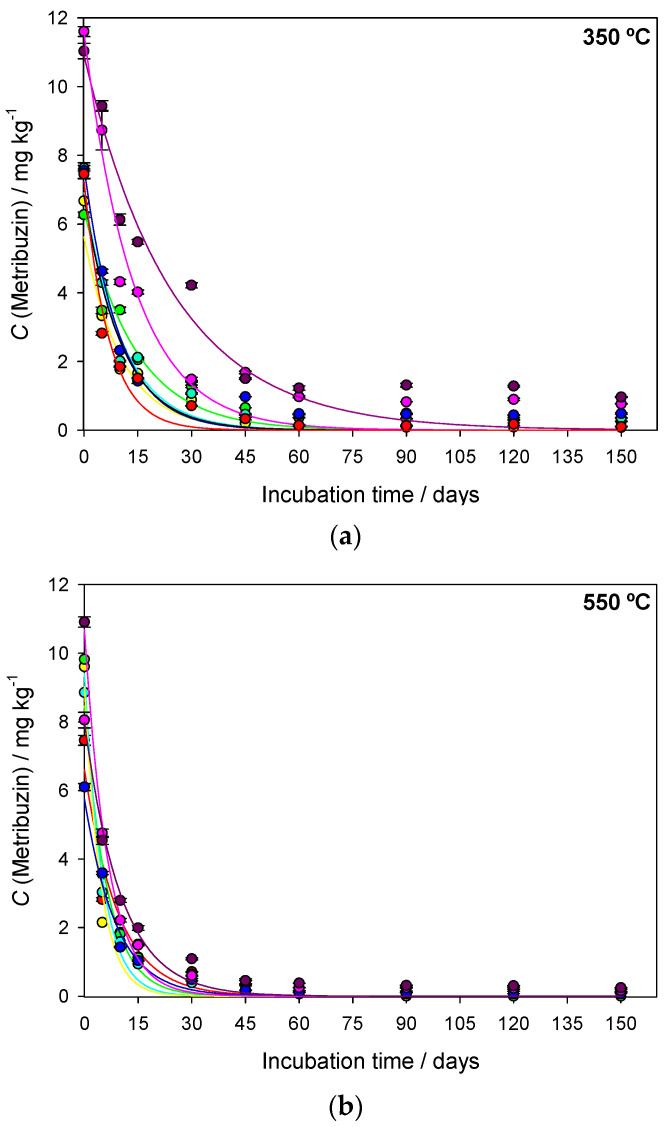
Concentration (*C*) of metribuzin in unamended soil and amended with different application rates of sugarcane straw biochar (BC) produced at different pyrolysis temperatures (**a**) 350 °C, (**b**) 550 °C, and (**c**) 750 °C. Degradation data were fitted to the kinetic model *C_t_* = *C*_0_ × *e*^−*k*x*t*^. Vertical bars represent the standard deviation of the means (*n* = 3).

**Figure 2 ijms-24-11154-f002:**
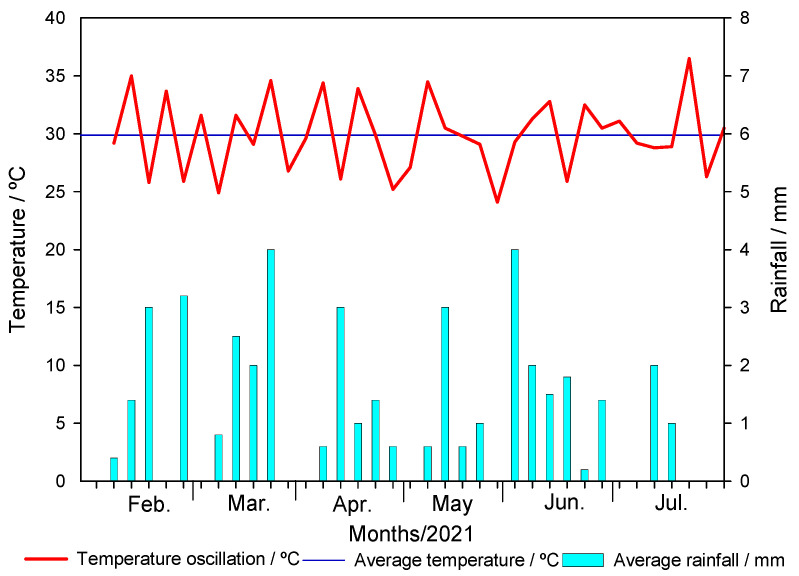
Temperature (°C) and precipitation (mm) recorded in the greenhouse in Viçosa, MG, Brazil, during the experimental period.

**Figure 3 ijms-24-11154-f003:**
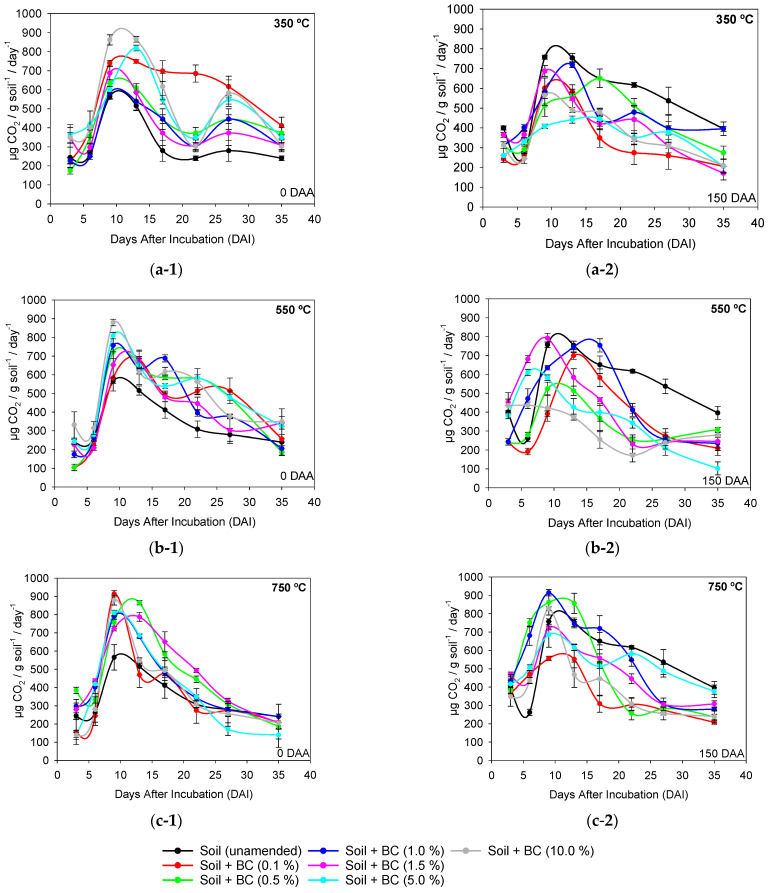
Microbiota respiration rate (C-CO_2_) at different incubation periods of soil amended and unamended with different application rates of sugarcane straw biochar produced at different pyrolysis temperatures (**a**) BC350 °C, (**b**) BC550 °C, (**c**), and BC750 °C, at 0 (**1**) and 150 (**2**) days after application (DAA) of metribuzin. The solid line represents the union of the dots that are the days after incubation (DAI) (3, 6, 9, 13, 17, 22, 27, and 35 days). Vertical bars at the symbols represent the standard deviation of the means (*n* = 4).

**Figure 4 ijms-24-11154-f004:**
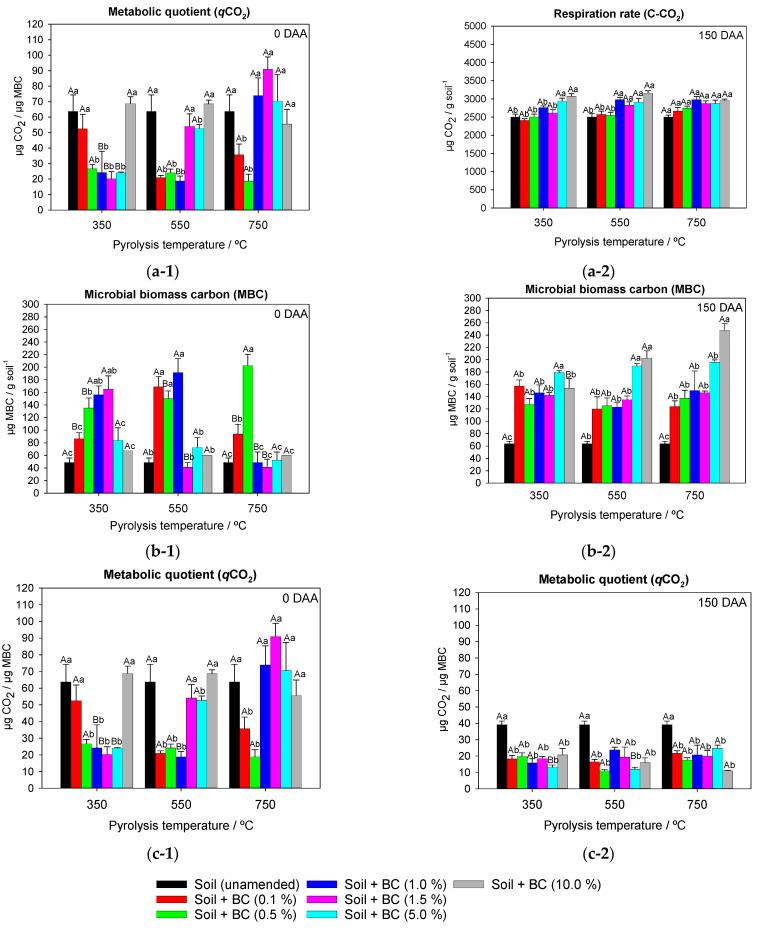
Respiration rate (C-CO_2_) (**a**), microbial biomass carbon (MBC) (**b**), and metabolic quotient (*q*CO_2_) (**c**) of soil microbiota amended and unamended with different application rates of sugarcane straw biochar produced at different pyrolysis temperatures BC350 °C, BC550 °C, and BC750 °C, at 0 (**1**) and 150 (**2**) days after application (DAA) of metribuzin. Same lowercase letters between application rates and same uppercase letters between pyrolysis temperatures do not differ by Tukey’s test (*p* < 0.05). Vertical bars at the symbols represent the standard deviation of the means (*n* = 4).

**Figure 5 ijms-24-11154-f005:**
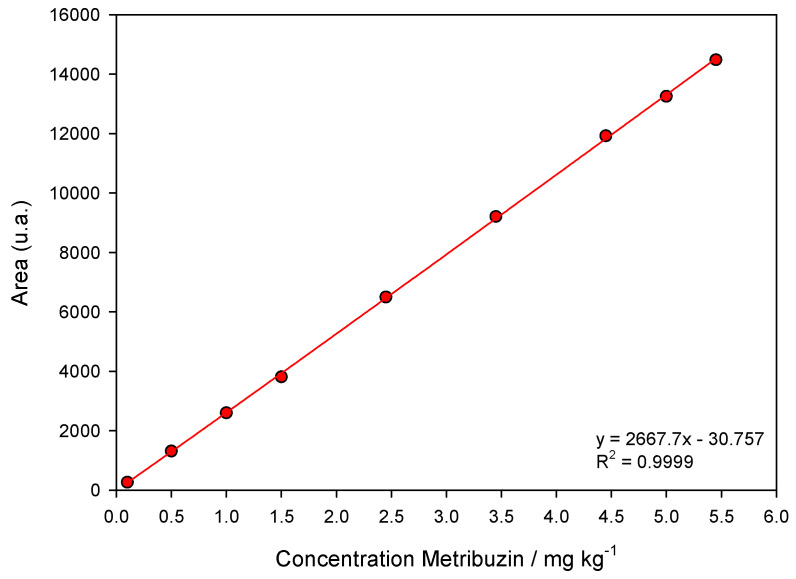
Linear chromatographic calibration curve of metribuzin. The points correspond to the mean (*n* = 3) of metribuzin concentrations (0.10, 0.50, 1.0, 1.50, 2.45, 3.45, 4.45, 5.0, and 5.45 mg kg^−1^). The line represents first-order linear model fit.

**Table 1 ijms-24-11154-t001:** Degradation parameters of metribuzin in unamended soil and amended with different application rates of sugarcane straw biochar (BC) produced at three different temperatures (350 °C, 550 °C, and 750 °C).

Pyrolysis Temperature/°C	Biochar Application Rate % (*w*/*w*)	*C* _0_	*k*	DT_50_	DT_90_	*p*-Value	*R* ^2^
		mg kg^−1^	days^−1^	Days	Days		
-	Unamended	7.22 ± 0.01 ^a^	0.094	7.37	24.49	<0.0001	0.989
350	0.1	5.61 ± 0.02	0.094	7.35	24.41	<0.0001	0.976
	0.5	6.39 ± 0.03	0.090	7.70	25.58	<0.0001	0.948
	1.0	6.91 ± 0.05	0.093	7.45	24.75	<0.0001	0.955
	1.5	7.64 ± 0.01	0.103	6.72	22.35	<0.0001	0.940
	5.0	11.55 ± 0.03	0.072	9.62	31.98	<0.0001	0.962
	10.0	10.89 ± 0.02	0.040	17.32	57.56	<0.0001	0.947
550	0.1	9.02 ± 0.05	0.220	3.14	10.46	<0.0001	0.966
	0.5	8.45 ± 0.08	0.156	4.44	14.76	<0.0001	0.991
	1.0	9.40 ± 0.07	0.199	3.46	15.57	<0.0001	0.991
	1.5	5.75 ± 0.01	0.134	5.17	17.18	<0.0001	0.987
	5.0	8.07 ± 0.01	0.138	5.02	16.68	<0.0001	0.991
	10.0	8.63 ± 0.02	0.114	6.08	20.19	<0.0001	0.983
750	0.1	5.22 ± 0.07	0.110	6.30	20.93	<0.0001	0.970
	0.5	6.85 ± 0.05	0.135	5.13	17.05	<0.0001	0.972
	1.0	8.18 ± 0.06	0.157	4.15	14.66	<0.0001	0.986
	1.5	6.41 ± 0.02	0.114	6.08	20.19	<0.0001	0.967
	5.0	7.15 ± 0.03	0.101	6.86	22.79	<0.0001	0.960
	10.0	6.04 ± 0.02	0.097	7.14	23.73	<0.0001	0.980

^a^ Average of the value of each parameter ± standard deviation of the mean (*n* = 3).

**Table 2 ijms-24-11154-t002:** Physicochemical attributes of the soil amended with sugarcane straw biochar and unamended soil.

Pyrolysis Temperature/°C	Application Rate % (*w*/*w*)	Chemical Attributes													
		pH	OC	P	K	Ca	Mg	H + Al	Zn	Fe	Mn	Cu	B	CEC	BS
		H_2_O	%	mg kg^−1^		mmol_c_ kg^−1^			mg kg^−1^					mmol_c_ kg^−1^	%
-	Unamended	5.5	1.2	1.3	77.0	15.9	5.4	33.0	3.0	129.8	91.0	3.9	0.1	23.3	41.0
350	0.1	5.5	1.2	1.5	97.0	16.0	5.7	33.3	2.9	129.6	99.1	3.8	0.1	24.2	40.0
	0.5	5.5	1.2	2.0	111.0	17.9	6.5	33.0	3.1	123.6	127.0	3.6	0.1	27.8	46.0
	1	5.8	1.2	3.3	125.0	17.5	6.8	26.4	2.8	148.1	130.0	3.7	0.1	29.3	52.0
	1.5	5.9	1.2	6.3	139.0	17.1	7.2	23.1	2.9	154.7	144.0	4.1	0.1	29.4	56.0
	5	6.8	1.2	10.0	240.0	17.7	8.3	13.3	2.9	234.4	155.0	3.8	0.1	36.7	73.0
	10	7.2	1.2	30.0	290.0	17.4	9.6	6.6	2.8	245.5	212.0	3.6	0.1	37.1	85.0
550	0.1	5.4	1.2	2.2	99.0	16.5	5.7	29.4	2.8	128.5	94.5	3.6	0.1	24.7	48.0
	0.5	5.6	1.2	2.7	132.0	16.2	5.8	29.7	3.1	157.4	97.9	4.1	0.1	24.8	45.0
	1	5.8	1.2	4.4	158.0	17.3	6.1	29.7	3.0	228.5	91.2	4.0	0.1	26.6	47.0
	1.5	5.9	1.2	8.7	161.0	17.8	5.8	19.8	2.8	266.5	157.0	3.4	0.1	25.2	56.0
	5	7.0	1.2	15.0	250.0	17.7	7.6	9.9	2.7	273.5	183.0	3.1	0.1	33.2	77.0
	10	7.3	1.3	33.0	340.0	18.1	8.4	3.3	2.9	297.5	202.0	3.6	0.1	38.5	90.0
750	0.1	5.4	1.2	2.9	108.0	16.8	5.6	33.0	2.7	135.0	96.6	3.6	0.1	25.2	43.0
	0.5	5.5	1.2	3.7	144.0	17.4	6.8	29.7	3.0	148.8	135.0	3.9	0.1	27.6	48.0
	1	5.8	1.2	7.8	178.0	17.8	7.0	29.7	2.8	147.6	122.0	4.0	0.1	29.4	49.0
	1.5	6.2	1.2	12.0	240.0	18.1	7.1	13.2	2.5	238.5	123.0	3.8	0.1	30.6	70.0
	5	7.2	1.3	55.0	500.0	19.7	9.8	3.3	2.9	267.5	177.0	3.7	0.1	39.3	92.0
	10	7.6	1.4	65.0	550.0	20.0	11.1	0.0	2.9	294.5	178.0	3.8	0.1	40.6	100.0
		**Physical attributes (g kg^−1^)**													
		**Sand**				**Silt**				**Clay**				**Texture class**	
Soil	Unamended	500				120				380				Sandy clay	

Source: from Mielke et al. [24] and Lab. Soil Analysis, Viçosa LTDA. Hydrogen potential (pH), organic carbon (OC), phosphorus (P), potassium (K), calcium (Ca), magnesium (Mg), potential acidity (H + Al), zinc (Zn), iron (Fe), manganese (Mn), copper (Cu), boron (B), cation exchange capacity (effective) (CEC), base saturation (BS).

**Table 3 ijms-24-11154-t003:** Selected properties of sugarcane straw biochar (BC) at different pyrolysis temperatures.

Pyrolysis Temperature/°C	pH	C	N	Ash	C/N	SSA
	H_2_O	%			-	m^2^ g^−1^
350	8.6	48.7	0.832	5.0	58.51	17
550	9.3	49.1	0.647	10.3	75.83	129
750	9.8	59.0	0.403	11.6	146.36	223

Source: Mielke et al. [24]. Temperature (T); hydrogen potential (pH); carbon (C); nitrogen (N); carbon/nitrogen ratio (C/N); specific surface area (SSA).

## Data Availability

Not applicable.

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
