# Peer review of "Pyrolysis Temperature vs. Application Rate of Biochar Amendments: Impacts on Soil Microbiota and Metribuzin Degradation"

_ijms, 2023, doi:10.3390/ijms241311154_

Round 1
Reviewer 1 Report
General comment:
The manuscript entitled “Pyrolysis temperature and application rate of sugarcane straw biochar influence soil microbiota and metribuzin degradation” is well written and structured research work and an interesting topic for readers from various fields including agronomy, pesticide science, environmental science, etc. The following revisions need to be considered before publication:
-Aside from this manuscript, these authors also published "Pyrolysis Temperature and Application Rate of Sugarcane Straw Biochar Influence Sorption and Desorption of Metribuzin and Soil Chemical Properties" in Processes (MDPI), https://doi.org/10.3390/pr10101924. Both of these studies appear to be part of one work, and it would be better if you published one comprehensive article by combining them. How do you feel about this?
-Introduction: In the introduction, you should discuss a number of key issues, such as herbicide adverse effects on the environment, human health, and non-target organisms (crops and microorganisms); metribuzin's site of action, mechanism of action/mode of action; as well as brief explanations of the biotic and abiotic degradation processes, especially photolysis.
-Introduction, Lines 30-31: You can use new appropriate citations such as Mehdizadeh et al. (2021).
Mehdizadeh, M.; Mushtaq, W.; Anusha Siddiqui, S.; Ayadi, S.; Kaur, P.; Yeboah, S.; Mazraedoost, S.; AL-Taey, D.K.A.; Tampubolon, K. Herbicide Residues in Agroecosystems: Fate, Detection, and Effect on Non-Target Plants. Rev. Agric. Sci. 2021, 9, 157–167.
-Introduction Lines 60-61: You stated: “The relationship between biochar addition in soil and metribuzin degradation is poorly reported in the scientific literature”. This statement does not seem to be true to me. This topic has been the subject of so many studies. Here are some examples:
https://doi.org/10.1080/03067319.2019.1597863
https://doi.org/10.3389%2Ffmicb.2022.1027284
-Results, Line 78: If applicable, please provide a chromatogram of the degradation of metribuzin.
-Materials and Methods, Line 359: Most studies conducted to evaluate pesticide degradation in agricultural soils use soil depths of 0-20 cm. What is the purpose of choosing a depth of 0-10 cm for this study? Give an explanation.
-Materials and Methods, Lines 378-379: The time used for oven drying should be mentioned.
-Materials and Methods, Line 418: Present the calibration curve and explain the calibration process.
Fair enough.
Author Response
Overall, the manuscript has been extensively edited (yellow color) to improve clarity. Reviewers comments are addressed in detail below. We thank the reviewers for their valuable comments and suggestions.
Reviewers' comments:
Review report 1
General comment:
The manuscript entitled “Pyrolysis temperature and application rate of sugarcane straw biochar influence soil microbiota and metribuzin degradation” is well written and structured research work and an interesting topic for readers from various fields including agronomy, pesticide science, environmental science, etc. The following revisions need to be considered before publication:
-Aside from this manuscript, these authors also published "Pyrolysis Temperature and Application Rate of Sugarcane Straw Biochar Influence Sorption and Desorption of Metribuzin and Soil Chemical Properties" in Processes (MDPI), https://doi.org/10.3390/pr10101924. Both of these studies appear to be part of one work, and it would be better if you published one comprehensive article by combining them. How do you feel about this?
These themes are linked, since the sorption capacity of a soil amended with biochar can influence the degradation of metribuzin in the soil. However, the choice to separate the themes is based on the idea of exploring more the relationship of the physicochemical characteristics of biochar produced at different pyrolysis temperatures in the sorption process of metribuzin in the soil. In a second moment, explore the relationship of sorption in the degradation of the herbicide in soil.
-Introduction: In the introduction, you should discuss a number of key issues, such as herbicide adverse effects on the environment, human health, and non-target organisms (crops and microorganisms); metribuzin's site of action, mechanism of action/mode of action; as well as brief explanations of the biotic and abiotic degradation processes, especially photolysis.
New explanations were added in the introduction.
-Introduction, Lines 30-31: You can use new appropriate citations such as Mehdizadeh et al. (2021).
Mehdizadeh, M.; Mushtaq, W.; Anusha Siddiqui, S.; Ayadi, S.; Kaur, P.; Yeboah, S.; Mazraedoost, S.; AL-Taey, D.K.A.; Tampubolon, K. Herbicide Residues in Agroecosystems: Fate, Detection, and Effect on Non-Target Plants. Rev. Agric. Sci. 2021, 9, 157–167.
The new citation was inserted in the text.
-Introduction Lines 60-61: You stated: “The relationship between biochar addition in soil and metribuzin degradation is poorly reported in the scientific literature”. This statement does not seem to be true to me. This topic has been the subject of so many studies. Here are some examples:
https://doi.org/10.1080/03067319.2019.1597863
https://doi.org/10.3389%2Ffmicb.2022.1027284
The sentence was rewritten and inserted in the correct way. When it comes to biochar and its dynamics in soil, there are different studies, however, the results are different depending on the biochar and rate used in the soil.
-Results, Line 78: If applicable, please provide a chromatogram of the degradation of metribuzin.
We chose not to insert it, since the degradation of metribuzin in soil is demonstrated in Figure 2, making the insertion of chromatograms repetitive. However, as requested by the reviewer, the calibration curve showing the recovery of metribuzin applied to soil will be added, according to the comment "Materials and Methods, Line 418: Present the calibration curve and explain the calibration process".
-Materials and Methods, Line 359: Most studies conducted to evaluate pesticide degradation in agricultural soils use soil depths of 0-20 cm. What is the purpose of choosing a depth of 0-10 cm for this study? Give an explanation.
The depth is very variable within degradation studies, as presented in the following papers:
https://doi.org/10.1016/j.hazadv.2022.100074
https://doi.org/10.1590/S0100-83582020380100034
https://doi.org/10.1016/j.apsoil.2020.103564
https://doi.org/10.1038/s41598-021-83052-z
https://doi.org/10.1016/j.enceco.2022.02.002
For our study, we chose to work with the arable layer normally adopted in practice. The superficial layer of 0-10 cm usually presents a greater amount of organic matter and microorganisms. In the case of a herbicide that was more sorbed by the action of biochar, a more superficial layer would be interesting for the study.
-Materials and Methods, Lines 378-379: The time used for oven drying should be mentioned.
Time was inserted into the methodology.
-Materials and Methods, Line 418: Present the calibration curve and explain the calibration process.
The calibration curve has been inserted into the text.
Reviewer 2 Report
Dear Authors,
Your manuscript seems interesting because it raises important issues related to herbicide degradation in soil. Unfortunately, it is insufficient, because I lack a clear summary. Such a summary can be written without research: "Therefore, the efficiency of biochar use in the soil depends on the purpose of its application."
In my opinion, the manuscript is good prepared. The layout of the articles became standard for research papers. However, I have a few comments and remarks:
Abstract: no conclusion
L. 68-70: I don't understand this sentence. It contradicts the cited publication [20].
Fig. 2,3, 4. Illegible. Engravings too small.
L. 118 and 478. Where does the "10.15 days" data come from? It's not in the tables?
The last paragraph in chapter 5 should be corrected. It adds nothing.
The language appears to be correct, but I don't feel qualified to judge about the English language and style.
Good luck!
Sincerely yours
Reviewer
Dear Authors,
Your manuscript seems interesting because it raises important issues related to herbicide degradation in soil. Unfortunately, it is insufficient, because I lack a clear summary. Such a summary can be written without research: "Therefore, the efficiency of biochar use in the soil depends on the purpose of its application."
In my opinion, the manuscript is good prepared. The layout of the articles became standard for research papers. However, I have a few comments and remarks:
Abstract: no conclusion
L. 68-70: I don't understand this sentence. It contradicts the cited publication [20].
Fig. 2,3, 4. Illegible. Engravings too small.
L. 118 and 478. Where does the "10.15 days" data come from? It's not in the tables?
The last paragraph in chapter 5 should be corrected. It adds nothing.
The language appears to be correct, but I don't feel qualified to judge about the English language and style.
Good luck!
Sincerely yours
Reviewer
Author Response
Review report 2
Dear Authors,
Your manuscript seems interesting because it raises important issues related to herbicide degradation in soil. Unfortunately, it is insufficient, because I lack a clear summary. Such a summary can be written without research: "Therefore, the efficiency of biochar use in the soil depends on the purpose of its application."
In my opinion, the manuscript is good prepared. The layout of the articles became standard for research papers. However, I have a few comments and remarks:
Abstract: no conclusion
The conclusion of the abstract is stated from line 20 to 22: The intensity of metribuzin degradation was dependent on the pyrolysis temperature and biochar application rate in the soil. The highest degradation of metribuzin was observed in soils amended with BC550 and BC750 when added at lower application rates (0.1 to 1.5%).
- 68-70: I don't understand this sentence. It contradicts the cited publication [20].
The sentence has been changed.
Fig. 2,3, 4. Illegible. Engravings too small.
The figures have been rearranged for better viewing.
- 118 and 478. Where does the "10.15 days" data come from? It's not in the tables?
The number is incorrect, it has been changed throughout the text. The correct number is 17.32 days.
The last paragraph in chapter 5 should be corrected. It adds nothing.
The last paragraph has been changed, as suggested by the reviewer.
The language appears to be correct, but I don't feel qualified to judge about the English language and style.
Ok.